# The Influence of Dietary Fibers on Physicochemical Properties of Acid Casein Processed Cheese Sauces Obtained with Whey Proteins and Coconut Oil or Anhydrous Milk Fat

**DOI:** 10.3390/foods10040759

**Published:** 2021-04-02

**Authors:** Jagoda O. Szafrańska, Siemowit Muszyński, Igor Tomasevic, Bartosz G. Sołowiej

**Affiliations:** 1Department of Milk Technology and Hydrocolloids, Faculty of Food Sciences and Biotechnology, University of Life Sciences in Lublin, Skromna 8, 20-704 Lublin, Poland; jagoda.szafranska@poczta.fm; 2Department of Biophysics, Faculty of Environmental Biology, University of Life Sciences in Lublin, Akademicka 13, 20-950 Lublin, Poland; siemowit.muszynski@up.lublin.pl; 3Department of Animal Source Food Technology, Faculty of Agriculture, University of Belgrade, Nemanjina 6, 11080 Belgrade, Serbia; tbigor@agrif.bg.ac.rs

**Keywords:** polysaccharide, computer vision system, texture, viscoelasticity, density, water activity

## Abstract

This study aimed to evaluate different fibers (acacia, bamboo, citrus or potato) on texture, rheological properties, color, density, and water activity of processed cheese sauces (PCS) based on acid casein, WPC80 and anhydrous milk fat or organic coconut oil. The interaction between the type of oil/fat, the fiber type and the fiber content was significant regarding almost all parameters studied. The computer vision system (CVS) showed that color changes of sauces could be noticeable by consumers. The main factor influencing the change in all products’ hardness was not fat/oil, but added fibers and their concentrations. The highest increase in hardness, adhesiveness and viscosity was observed in products with potato fiber. The value of storage modulus (G′) was higher than the loss modulus (G″) and tan (δ) < 1 for all samples. Different fibers and their amounts did not influence the water activity of cheese sauces obtained with organic coconut oil (OCO) or anhydrous milk fat (AMF).

## 1. Introduction

At the end of the year 2018, Innova Market Insights, a company tracking new food and beverage products on present and future trends, published a forecast for 2019, “Top 10 Trends for 2019”. One of the presented vital points was “A fresh look at fiber”. Analytics suggested that consumers growing interest in fiber will cause the manufacturing of new fiber applications in the next few years [1]. Consumers have started to pay attention to the composition of food products they buy and are increasingly willing to reach for products with higher dietary fiber content. According to several systematic reviews and meta-analyses, clinical research and observational studies performed for almost 40 years show the advantages of having a healthy diet by consuming at least 25 g to 29 g or more of dietary fiber a day [2]. Fibers can also be incorporated into dairy products to meet such health requirements. This type of innovation can increase fiber-rich foods availability and people’s understanding of their advantages to their health.

Most of the dairy-based foods are considered to belong to the high-moisture group of products. For this type of application, soluble fibers generally work excellently. Processed cheese sauce is one of the food products a part of the described group with high water activity. Because the formulations do not have strict standards or legal definitions, they may be made up of a number of different dairy or non-dairy ingredients, including fiber [3]. Commercial cheese sauces are a mixture of various ingredients. The typical base of cheese sauces includes cheese solids, different flavor systems and fat sources (including animal fat: butter, buttermilk and lard or/and vegetable oils: palm, soy, or canola). In addition, texture additives and fillers [4]. For the processed cheese sauce production described and tested in this article, semi-finished products with additional health-promoting properties were used. The main ingredient of the described formula is whey protein concentrate (WPC80). It is a rich source of amino acids and contains a high concentration of essential amino acids [5]. Additionally, another ingredient of the described formula, acid casein, has water-binding properties [6]. Coconut oil (OCO) contains medium-chain fatty acids, which correspond to 64% of the total fat. According to the findings, medium-chain saturated oil may help reduce intermuscular and abdominal fat more effectively than long-chain unsaturated oil [7]. It was found that some nutritional benefits of OCO were due to its chemical composition. Numerous studies have been conducted to describe the possible health-promoting effects of coconut oil, e.g., a positive cardioprotective/hypocholesterolemic effect, anticancer, antidiabetic or hepatoprotective activity [8]. In turn, tests conducted to determine the properties of anhydrous milk fat (AMF) show that consumption of AMF causes a decrease in the concentration of low-density lipoprotein (LDL) and decreased amount of liver total cholesterol and cholesterol esters [9]. Traditional cheese sauces can contain around 3% of emulsifying salts, which are commonly considered harmful to health in higher doses. Sołowiej et al. [10] reported that replacement of typical emulsifying salts (phosphates) could be possible due to the use of whey proteins [10]. In the designed product formulation, the amount of emulsifying salts has been reduced to just 0.8%, due to WPC80.

Another innovation in the presented formula is the addition of dietary fibers. They can play an important role in preventing many chronic diseases, such as diabetes, obesity, atherosclerosis or heart diseases [11].

Dietary fiber is a part of plant material which includes cellulose, noncellulosic polysaccharides (hemicellulose, pectic substances, gums, mucilages) and also a non-carbohydrate component: lignin. In our study, four types of dietary fiber (acacia, bamboo, citrus and potato fiber) were used. Dietary fiber is easy to incorporate, stabilizes the dairy product and maintains its creamy and smooth final texture [12]. The first attempt to obtain a new type of processed cheese sauce based on rapeseed oil with the addition of fibers was described in 2020. The received textural and rheological results provided impressive knowledge about improvements in the properties mentioned above. Most prepared samples had lower hardness. This research indicated that fibers can be potentially used in processed cheese sauces [13]. In this article, we expanded the research scope with additional analyses of color using a new type of method for cheese sauces color assessment—the computer vision system (CVS), measurements of the tan (δ), yield stress, water activity and density of tested cheese sauces. Moreover, to the authors’ knowledge, there is no research concerning the effect of dietary fibers on acid casein processed cheese sauces obtained with coconut oil or anhydrous milk fat, thus, to obtain PCS in this research, the sources mentioned above were used.

Trans and saturated fatty acids are related to the possibility of developing cardiovascular diseases. However, intake of anhydrous milk fat (AMF) is correlated with health-promoting effects [9]. It has a minimum of 99.8% milk fat and no more than 0.1% moisture [14]. Due to certain medium-chain fatty acids, organic coconut oil (OCO) is perceived as healthy because of its capability to be quickly absorbed by the digestive system, causing satiety and preventing fat storage [15]. Additionally, the whey protein concentrate (WPC80) used to prepare our product also has health-promoting properties [16]. It is characterized by antibacterial, antitumor [17] and antifungal properties [18]. Acacia fiber is an emulsifier with low-viscosity and high solubility in cold aqueous solutions [19]. Bamboo fiber contains over 70% cellulose (insoluble), 10% lignin (insoluble) and only 12% hemicellulose (soluble) [20]. The majority of the composition of citrus fiber is pectin (around 40%), hemicellulose (10%) and cellulose (15%) [21]. Commercial potato fiber is composed of lignin (2%), cellulose (over 23%), pectin and hemicellulose (45%) and total content of non-starch polysaccharides (59%, of which 23% is soluble and 32% is insoluble) [22].

Thus, this study aimed to test the influence of different dietary fibers (acacia, bamboo, citrus or potato) on rheological and texture properties, color, density and processed cheese sauces’ water activity. Our experiment focused on obtaining processed cheese sauce based on whey protein concentrate (WPC80) and acid casein with the addition of various dietary fibers that previously have never been used in this type of product, which was obtained with coconut oil or anhydrous milk fat.

## 2. Materials and Methods

### 2.1. Materials

In the production of processed cheese sauces (PCS), the following raw materials were used: acid casein (AC, 92.1% proteins, PPHU Polsero, Sokołów Podlaski, Poland), whey protein concentrate (WPC80, 76.8% proteins, Milkiland EU Ltd., Warsaw, Poland), organic coconut oil (OCO, Bio Planete, Lommatzsch, Germany), anhydrous milk fat (AMF, 99.8% fat content, Mlekovita, Wysokie Mazowieckie, Poland), acacia fiber (AF) (Nexira, Chemin de Croisset, France), bamboo fiber (BF) (Beneo-Orafti SA, Belgium, citrus fiber (CF) (Roeper GmbH, Hamburg, Germany), potato fiber (PF) (Lyckeby, Starch AB, Fjälkinge, Sweden), lactic acid, disodium phosphate and sodium hydroxide (PPH POCH, Gliwice, Poland).

A series of samples of processed cheese sauces were prepared to carry out a series of individual tests described in the Material and Methods subsection. In general, 32 samples of tested PCS were prepared in the appropriate number of repetitions needed to perform the tests with the methods described below.

### 2.2. Determination of Proteins

According to AOAC (Association of Official Analytical Chemists) requirements, the protein content of acid casein (92.1%) and whey protein concentrate (76.8%) was determined by measuring the nitrogen using Kjeldahl methods and calculating protein as N·6.38 [23].

### 2.3. Preparation of Processed Cheese Sauces

WPC80 (6%, *w*/*w*) and AC (6%, *w*/*w*) were mixed in distilled water using a magnetic stirrer (Heidolph MR 3002S, Schwabach, Germany) (300 rpm, temp. 21 °C). Then, 10% of OCO or AMF (constant value for each sample, melted in 45 °C), followed by 1, 2, 3 or 4% of four fibers: AF, BF, CF or PF were added and constituents were placed in a vessel and mixed using the H500 homogenizer (Pol-Eko Aparatura, Wodzisław Śląski, Poland) for 2 min at 10,000 rpm. Disodium phosphate (0.8%, *w*/*w*) was added, pH was adjusted to 5.8 using sodium hydroxide or lactic acid (2 M), the mixture was immersed in 80 °C water bath and the ingredients were mixed at 10,000 rpm for 10 min according to Szafrańska et al. [24].

### 2.4. Penetration Test

The TA-XT2i Texture Analyzer (Stable Micro Systems, Godalming, Surrey, UK) was used to perform all measurements, which followed the protocol defined by Szafrańska et al. [24]. A 15 mm diameter cylindrical probe was used to penetrate processed cheese sauces to a depth of 28 mm. The rate of penetration was 1 mm/s. Texture Expert software was used to determine the hardness and adhesiveness of processed cheese sauces. Five measurements were taken for each of the three replicates.

### 2.5. Viscosity

A Brookfield DV II+ rotational rheometer (Brookfield Engineering Laboratories, Stoughton, MA, USA) fitted with a Helipath Stand and T-bar spindle D was used to examine the apparent viscosity of processed cheese sauces. According to the method of Szafrańska et al. [24], three measurements were carried out at 21 °C with a spindle velocity of 0.5 rpm.

### 2.6. Viscoelastic Properties

According to Szafrańska et al. [24], storage (G′) and loss (G′′) moduli, tan (δ) and yield stress of cheese sauces were measured using serrated plates (PU40X SW1382 SS and PLS40X S2222 SS, at the plate—plate configuration) on a Kinexus lab + rheometer (Malvern Panalytical, Cambridge, UK). The Kinexus Malvern software—rSpace—was used to record the results of the measurements (three repetitions), which were made at 21 °C.

### 2.7. Colorimetric Measurements—Computer Vision System (CVS)

Colorimetric measurements (three repetitions) of obtained processed cheese sauce (PCS) was described with parameters L* (0–100, estimation of lightness), a* (red-green) and b* (yellow-blue) using a computer vision system (CVS) with the use of Sony Alpha DSLR-A200 digital camera (10.2 Megapixel CCD sensor) according to the method described by Tomasevic et al. [25]. According to the National Bureau of Standard reference scale, the notable differences could be described, which identified that such changes are noticeable to the human eye.

The total color difference was calculated using the formula:(1)$$\Delta E=\sqrt{{\left(a_{1}-a_{2}\right)}^{2}+{\left(b_{1}-b_{2}\right)}^{2}+{\left(L_{1}-L_{2}\right)}^{2}}$$

The ΔE* values were converted into National Bureau of Standards (NBS) units by the equation [26]:(2)NBS units=ΔE×0.92

### 2.8. Water Activity

The Aqua Lab 3 TE water activity meter (Decagon Devices Inc., Pullman, WA, USA) was used to measure water activity (aw) with a precision of +/− 0.001 of an a_w_ unit. The Rotronic humidity standard was used to calibrate the apparatus prior to measurement (95% HR). According to Szafrańska et al. [24], measurements were carried out in five repetitions at a temperature of 22 °C. Two outliers were reported as defective in each sample and were removed from further analysis.

### 2.9. Density

At a temperature of 22 °C, a gas pycnometer (AccuPyc 1330; Micromeritics, Norcross, GA, USA) was used to determine density [24]. The analysis was performed in three repetitions of each tested product.

### 2.10. Statistical Analysis

The data were analyzed using a 2 × 4 × 4 factorial arrangement with a type of oil/fat (AFM or OCO), fiber type (AF, BF, CF, PF) and fiber concentration (1%, 2%, 3%, 4%) as the factors and their interactions (a three-way ANOVA). The Shapiro–Wilk test was applied to verify which variables had a normal distribution; the Levene test examined homogeneity of variances. As all the data show normal distribution, treatment means were separated using the Duncan’s test when significant differences were found among treatments. For all tests, a *p*-value < 0.05 and a confidence interval of 95% were established. PCA (principal component analysis) was also applied to the average values of physical properties (hardness, adhesiveness, viscosity, yield stress, G′, G″, tan (δ)) of all samples studied. The Kaiser–Meyer–Olkin measure (KMO) and Bartlett test of sphericity were used to determine the PCA’s significance. All calculations were carried out with Statistica software (v. 13.3, TIBCO Soft-ware Inc., Palo Alto, CA, USA).

## 3. Results and Discussion

To comparatively analyze the influence of the type of oil/fat (AFM or OCO), fiber type (AF, BF, CF, PF) and fiber concentration (1%, 2%, 3%, 4%), a three-factor ANOVA with interactions was performed (Table 1). The outcomes reported that the interaction between the type of fat/oil, the fiber type and the fiber content was also significant for almost all parameters studied. It means that the effect of the type of fat/oil on studied parameters depended on the fiber type and its concentration in the system. Only for water activity, the effect of the type of fat/oil depended only on fiber content, irrespective of its type.

### 3.1. Penetration Test

The effect of different fiber types and concentration on PCS texture attributives obtained-based on acid casein with AMF, or OCO is presented in Figure 1 and Figure 2. Generally, the final products’ hardness depended on the amount and type of added fiber and fat/oil type. The hardness of PCS with the addition of AF and BF decreased along with an increased amount of fiber (*p* < 0.05). In both formulas with OCO and AMF, the same tendency was observed. The greatest hardness characterized PCS with 4% PF + AMF (0.863 N), while the products with the lowest hardness were obtained with AMF/OCO with 3 and 4% addition of AF/BF. In addition, 1% PF + OCO/AMF (*p* < 0.05) (Figure 1).

Understanding the structure of fibers and fat/oil used in research is important to describe better and understand the differences between hardness values of final products. The primary classification, commonly used, describes fiber as a plant-based nutrient containing soluble and insoluble parts [27]. Both types of dietary fibers have similar physical properties, including the ability to bind water [28]. However, polysaccharides classified as capable of dissolving in water can be quite variable in their actual solubility [29]. The part that is considered soluble components of DF are hemicelluloses, pectins, gums and mucilage. Lignin, cellulose and resistant starch are described as insoluble fractions. Many of the mentioned polymer types can be classified as soluble or insoluble depending on the plant source and degree of post-harvest processing [30].

Results suggested that the main factor influencing the change in all products’ hardness was not fat/oil, but added fibers and their concentrations. In each sample, hardness was increased in sauces with addition of CF + OCO (from 0.08 to 0.5 N) CF + AMF (from 0.1 to 0.5 N) and PF + OCO (from 0.07 to 0.5 N) and PF + AMF (from 0.06 to 0.9 N). In each case, about half of the fiber composition contains soluble fractions. Due to better binding capacity with water, the final processed sauce structure became more compact, thus, harder. In 2020, we conducted a series of experiments examining the effect of rapeseed oil and various fibers on the cheese sauce structure. We observed an increase in hardness values in products with the addition of citrus fiber, which is similar to the described results in products with CF + OCO/AMF [13]. We did not notice an increase in the value mentioned above in sauces with potato fiber addition, suggesting that oil/fat addition has a great impact on product hardness. To the best of our knowledge, no research data concern dietary fibers’ effect on PCS-based on acid casein and coconut oil or milk fat. Akalın et al. [31] tested the structure of ice creams with the addition of dietary fiber. They added different fibers (apple, orange, oat, bamboo and wheat) and noticed that each product was characterized with good hardness, except for the product supplemented with bamboo fiber. They also noticed that the highest hardness values were observed in ice cream with orange fiber [31]. In both examples, the gelling properties and water-holding capacity provided a better and more compact composition of final products. There is no study describing incorporating potato fiber to different dairy products, but few attempts to test it in other food products—bread enriched with potato fiber. Kaack et al. [32] noticed the increase in hardness of bread, which can be connected with protein hydration and, consequently, matrix plasticization [32].

Among all examined samples, the highest level of adhesiveness characterized the sauce with the addition of 4% PF + OCO (437 J). PCS with 4% PF + AMF (404 J) and 4% CF + OCO (415 J) were characterized with results very close to the highest value noted (*p* < 0.05) (Figure 2). The adhesiveness of products with AF + OCO and BF + OCO/AMF was systematically decreased as the added fiber percentage increased. In each case, the value of adhesiveness was lower than 300 J. Sanchez et al. [33] analyzing the available literature on the properties and features of acacia gum noticed that as the concentration of acacia gum (acacia fiber) in dispersions increased from 3% to 10%, the amount of water bound decreased from 1.2 g to 0.6 g/g of gum [33]. Our findings show that as the concentration of these fiber macromolecules increased, some self-association (i.e., aggregation) occurred. It leads to decreased water accessibility and bound water release from acacia fiber macromolecules during the aggregation process. The noticeable phase separation observed in our study, particularly with the addition of AF, could also be linked with decreased adhesiveness. This decrease in products’ adhesiveness and a growing amount of added fibers has also been noticed in sauce with bamboo fiber. Differences between these two products were observed between samples with different oil/fat. Product with BF + OCO presented the same results as with the addition of AMF, which was not noticed in PCS with AF. In addition, in our research from 2020 on PCS with the addition of DF and rapeseed oil, we noticed a different tendency of adhesiveness between the different percentages of added fibers. In products with CF and PF, the value systematically decreased along with an increase in added fibers, suggesting that oil/fat added to cheese sauce was the main factor influencing its properties [13]. Animal fat composition depends on many factors, including diet and seasoning. Vegetable fats like coconut oil are less variable. Anhydrous milk fat is a mixture of triacylglycerols, contains at least 60 different fatty acids and has unique thermal and chemical properties [34]. The physical properties of fat that influence food features containing them are mainly concerned with the phase changes that occur during these state transitions, such as solid to solid, liquid to solid or solid to liquid [35]. Milk fat has its source in the globule, predominantly in the form of spherical globules enclosed by a membrane made up of protein and phospholipid. Anhydrous milk fat is the final product after the membrane content has been almost completely eliminated [36]. AMF contains a small amount of phospholipids (around 0.01%) [37]. This disproportion may be related to how the adhesiveness changes in individual sauces. Rich fat globule membrane that is removed from AMF has a great impact on adhesiveness. Le et al. [38] prepared yogurts enriched with milk fat globules and reported increased water-holding capacity and stronger adhesiveness [38]. Despite the idea of incorporating fibers including bamboo [31], acacia [39] or citrus [40] to dairy products like ice-cream, mozzarella cheese or yogurts, to the best of our knowledge, there is lack of research focusing on parameters including adhesiveness of PCS with addition of fibers.

### 3.2. Viscosity

Viscosity parameter relating to dietary fiber as a component of food product, when mixed with fluids, refers to the ability of some polysaccharides to thicken or form gels as a result of physical entanglements among the polysaccharide constituents within the prepared fluid or solution [41]. Figure 3 illustrates the viscosity of individual products based on AC and WPC80 with OCO or AMF and the addition of the various amounts of different fibers. Generally, the viscosity of PF + OCO/AMF and CF + OCO/AMF cheese sauces increased along with higher fiber concentration. Simultaneously, in sample with BF + OCO/AMF, the inverse trend was observed, suggesting that the added fiber’s structure has exerted the opposite effect on the final product completely. In sauces with AF + OCO/AMF, the viscosity value remained at a similar level. The viscosity of 4% PF + AMF was the highest (35,400 Pa·s).

Acacia fiber contains a molecular structure that is strongly branched and has a low molecular weight may cause a change in the value of viscosity and high water solubility properties [42]. Our products had a fluid structure after adding acacia fiber and in some cases, we observed delamination in the final product. In addition, bamboo fiber is characterized by high water holding capacity. A significant reduction in viscosity with an increase in the content of fiber was observed. The composition of citrus fiber might cause a systematic increase in viscosity in final products. Carbohydrates, including pectin and cellulose, represent the majority of CF. Due to the pectin components’ acidic nature, apparent viscosity and gelling properties make this fiber perfect for food applications [21]. The highest values of viscosity for the sauce with potato fiber may be a result of the properties described in Section 3.1, i.e., a higher amount of insoluble fraction and low viscosity [22]. It can bind and reduce the free water in sauce and restrict the food system’s movement and, because of it, an increase in the viscosity was noted.

In addition, the differences between a product with OCO or AMF were observed. Sauces with OCO in few cases had higher viscosity value than products with AMF (Figure 3). Basically, interactions between the fat-solid component involve the coating of particles by fat. Fat penetrates the solid matrix and causes viscosity effects [35]. It may suggest that OCO blends better in our sauce base structure and individual products obtained higher viscosity values than those with AMF addition. It was also observed that the viscosity of the AMF is related to factors such as the shear rate or temperature. Viscosity at a higher shear rate (1440 s^−1^) was lower compared to 90 s^−1^. In addition, the temperature used during the experiment (45 °C to 17.5 °C) impacted the examined sample. The described feature increases to a certain point and then progressively decreases [43]. The pH of the product can also influence on the viscosity. Adsorption of whey proteins used to prepare the sauces on fat globule surfaces of coconut oil depends on pH value. It was observed that the volume of whey proteins adsorbed on fat at pH 5.0 in coconut emulsions was more remarkable in comparison to other tested pH (3.0, 7.0 and 9.0) [44]. During our products’ preparation, we applied pH 5.8, which seemed to work well with the OCO, compared to samples with AMF.

### 3.3. Viscoelastic Properties

Generally, fluids and semisolids that exhibit yield stress are characterized by a structural network, the strength of which is dependent on the type of interactions between each molecule that composes it. Furthermore, the structure of the dispersed phase can have an impact on the structural network of fluids. Yield stress determines the essential features of cheese sauces. The rheological property of liquid and solid materials describing the material structure’s strength is defined as the minimum stress necessary to make a tested material flow [45]. Table 1 presents the value (Pa) of yield stress (yield point) in an instance when the product cannot sustain more elastic deformation and start flowing. In most of the samples, the value of yield stress increased along with the amount of added fiber. In contrast, a decrease in the yield stress parameter and a higher amount of added fiber was observed in sauces with OCO/AMF + AF. In sauces with AMF + BF, the testing parameter values increased in product with the addition of 1–3% fiber and were much lower regarding 4% BF. In addition, the values of yield stress—both in samples with the addition of AF and BF—were much lower than other tested samples.

Whey—due to its structural elements, the occurring proteins including β-lactoglobulin, α-lactalbumin and immunoglobulins—can create three-dimensional structures held together by disulfide bridges [46]. The described components, α-lactoglobulin and β-lactalbumin, can stabilize emulsions because of their capability to adsorb oil–water interfaces [47]. When analyzing the obtained results, it can be concluded that semisolid fluids such as cheese sauce presented shear-thinning behavior. Regarding this, there are a few hypotheses: One of them explains that, due to large molecular chains that randomly tumble and with the sizeable hydrodynamic radius, it could significantly affect the resistance to fluid flow under low shear. The increasing shear rate value proves that these large molecular chains align themselves in the direction of shear force and, during that process, they become resistant to shear force decrease [48]. We also observed that the amount of stress that the product needs in order to experience a permanent deformation was the highest in sauces with AMF + 4% CF, AMF + 3 and 4% PF and OCO + 3 and 4% PF. The value they reached was higher than 1000 Pa. These results were correlated with hardness measurements. The described sauces also have the most outstanding hardness values, suggesting that they had the most compact structure. Yield stress is also an essential product characteristic that determines its texture and consumer sensory perception during its use and application [45].

In Table 2, the influence of dietary fiber’s concentration on PCS with the addition of OCO or AMF on storage (G′) and loss (G″) moduli were presented. When G′ exceeds G″, it means that tested material has a structure associated with yield stress. We can observe in our product that the value of G′ was higher than G″ at infinitely low frequencies.

The measurement results for G′ and G″ were very diverse. Regarding tested sauces with PF + AMF/OCO (2–3%) and BF + AMF (1–4%), the value of storage and loss moduli increased along with the amount of added fiber. The prepared samples behave as an elastic solid, although this is not ideal because some accumulated mechanical energy is dissipated. The presented results may prove that tested sauces have strain-dependent networks. They tend to flow differently than true gels and they are more solution-like under comparatively minor stresses [49]. In samples with AF + AMF (1–4%) and BF + OCO (1–3%), a decrease in the described features along with the amount of added fibers was observed. The values of the storage (G′) modulus were higher than the loss (G″) modulus in all tested PCS. It suggests that the prepared samples exhibited elastic properties during the whole measurement. The presented results are consistent with the data obtained during the tests of cheese sauces with the same fibers based on rapeseed oil. The G′ values exhibited higher values in sauces with bamboo and potato fibers. Our research from 2020 suggests that this may be caused because PF and BF consist of fewer water-soluble fractions [13].

In addition, a relationship in decreasing and increasing of values between storage modulus and hardness was noted. Regarding PCS with AMF, sauces with 1% of AF and PF had the highest G′ and G″ values, implying that the gel structure of the PCS was the strongest and formed a more flexible system than products with a more significant addition of fibers. In other products, this tendency was reversed. The highest values of loss and storage moduli were observed in 3% AF + OCO sauce (60,086 Pa) and the lowest in 2% AF + OCO (0.70 Pa) and are consistent with our observations. This could be because acacia fiber has a higher amount of water-soluble fractions. The higher content of the insoluble fraction of the used fiber most likely served as a gel filler. This can cause the reinforcement of the structure.

Another tested feature was the tan (δ) of tested PCS. The loss tangent (tan (δ)) was described as follows:(3)$$\tan\left(\delta \right)=G^{\prime \prime }/G^{\prime }$$

It is defined as a phase angle between the viscous and elastic components of material behavior [50]. When G′ represents higher values than G″, it means that tan (δ) <1. Such values indicate that measured samples have more elastic than viscous properties. Values of tan (δ) for each of the tested products are lower than one, which means that obtained PCS have elastic properties. Additionally, we noted that the products with AMF and AF, CF, or PF generally have lower tan (δ) values than other tested sauces samples. Lopez et al. [51] observed that the anhydrous milk fat melts at 40–41 °C [51]. Melted fat may partially fill the inter-protein spaces in a product based on WPC and the remaining part of it increases the volume of the cheese sauce sample, causing a boost in the value of G′ and G″ modules and simultaneously a reduction in the value of tan (δ). Additionally, the temperature used during the homogenization process (80 °C) may be connected with the process of gel formation. When lactose content in concentrates and isolates is low, their gelation temperature is shifted during whey protein’s denaturation [52].

### 3.4. Principal Component Analysis (PCA)

A PCA was applied to compare the samples of examined processed cheese sauces and analyze the variability in their physical properties parameters. The Kaiser–Meyer–Olkin (KMO) measure of sampling adequacy was 0.637 and the Bartlett test of sphericity (Chi^2 value 183.147) reached statistical significance (*p*-value < 0.001), providing the basis for the application of PCA analysis. The results showed that a high concentration of CF and PF fiber type was the factor that most affected most of the physical parameters, irrespective of the oil/fat type. The map obtained after PCA was applied is shown in Figure 4. The first two principal components explained a high amount of the variance (77.09%), from which the first component explained most of the data variability (46.08%). It was negatively associated with hardness, viscosity, adhesiveness and yield stress (Figure 4, inset), although the correlation matrix indicated that the information provided by these four parameters was very similar. This element was primarily responsible for the separation of the examined processed cheese sauces with the high concentrations (3 or 4%) of CF and PF fiber type (Figure 4); in particular, both OCO and AMF samples with 4% of PF fiber were clearly separated from the rest and located on the negative part of the first dimension. The latter two cheese sauce samples were characterized with the highest values of yield stress, viscosity, hardness and adhesiveness (Figure 1, Figure 2 and Figure 3, Table 1). The second component described a lower amount of the variance (31.01%) and was highly negatively correlated with G′ and G″ (Figure 4, inset). The majority of samples were distributed on the positive side and only three OCO samples (1% and 4% PF and 3% AF) and 4% PF + AMF sample fell in the negative part of this component. Mostly, a 3% AF + OCO sample was significantly separated from the rest. Indeed, this sample was characterized by an extremely high value of G′ (Table 1).

### 3.5. Colorimetric Measurements

Table 3 presents the color analysis results obtained using the computer vision system (CVS). One of the most prominent variables affecting consumer preference is color. Therefore, it is critical to prepare a product that is closest in appearance to consumers’ preferences. As opposed to other approaches, CVS color measures are much closer to the true color of the samples. Furthermore, the obtained color shades are more saturated than those obtained using conventional colorimetric techniques [25].

The intensity of PCS color was significantly different between tested samples (*p* < 0.05) and it could be described as creamy white to creamy yellow because b* in the products with OCO addition and AF, BF, CF have lower values than a product with the addition of the same fiber but-based on AMF.

Only in the sauce with PF(1–3%) + OCO, values of b* were higher than PF + AMF. Of all tested samples, sauces with the addition of AMF were identified by the highest b* values and were the most yellow-tinted in comparison to the other samples. All tested samples had positive values of a* parameter, which indicated a slight red hue. The brightness (L*) of each of the measured samples ranged from 79 to about 88. Statistically significant differences for the L* were observed between samples in each product group (*p* < 0.05). In terms of L* value in cheese sauces obtained based on different oil/fat, it can be observed that they do not affect this parameter. Still, among different fibers, significant differences were noted (*p* < 0.05). The lowest L* was observed in a group of sauces with the addition of PF. Wadhwani [53] noticed that cheese color’s prime determination has light scattered by milk fat globules [53]. We did not observe the same results in every PCS sample due to used fibers, but products with AMF addition had more yellow tones than most OCO products. Olsson, Håkansson, Purhagen and Wendin [54] tested influence of emulsion intensity on the textural characteristics of full-fat mayonnaise. They suggested another factor that can cause changes in the color of the final product. They noticed that reducing droplet size during the homogenization process leads to a whiter of the final product [54]. We also noticed that used fibers did not affect the color of our product. The bamboo fiber used to prepare samples had a white color, whereas acacia and citrus fiber had a creamy white shade. Only potato fiber was characterized by visible light-brown particles, the color of which was not so diametrically visible in the sauce’s final color.

For a comprehensive description of the product’s color, we provide information about another attribute: chroma and hue angle. All the tested products had different hue angles. The lowest value characterized samples with OCO + 1–3% PF (*p* < 0.05). The measurements showed that sauces’ colors were between 5–20, which corresponds to the orange color. The other appearance parameter describing the developed product is chroma. For a sauce obtained with OCO’s addition, chroma’s highest content was observed among samples with 1–3% PF, with no significant difference between them (*p* > 0.05). Meanwhile, these cheese sauces were characterized by the highest b* and a* representing more green and yellow final product tones. PCS obtained-based on AC + AMF had the highest content of chroma in products with the addition of AF, BF and CF (*p* < 0.05).

The color of individual products was influenced by both the addition of various oils and dietary fibers (Figure 5). PCS obtained based on AC + AMF have more yellow tones than sauces with AC + OCO. This could be related to the color of milk fat. It has natural yellow tones connected with carotenoids, vitamin A and other pigments [55].

Color distinctions represented by ∆E values between 00.00 and 1.5, according to Inami et al. [26], are incredibly slight; consumers can notice slight changes between 1.5–6.0 [26]. Values of ∆E above 6 prove the significant influence of the dietary fiber and fat source on PCS’s color difference between prepared samples. Table 4 presents the results of color intensity measurements of PCS expressed by NBS units (∆E × 0.92). A comparison of tested samples between each other showed that most of them characterized by NBS higher than 1.5. This means that changes can be noticeable. The most visible changes occur in CF + OCO compared to AF + AMF and AF + OCO compared to AF + AMF. These results follow the results discussed above. Dietary fibers from various sources have similar colors. In smaller amounts, their addition to sauce does not change the final products’ shade significantly as fat used in production. The only amount of added fibers is the factor that influenced the color of the products. More fiber increased the overall color of the PCS.

### 3.6. Density and Water Activity

Results of density and water activity of PCS have been presented in Table 5. Generally, fibers and fat/oil affected the density of processed cheese sauces. Fat/oil can form emulsions, which are dispersions of a fat or oil into water (or water into oil/fat). The emulsification process of fat into food products gives them unique texture qualities [56].

Density decreases when fats are changing their form from solid to liquid. During the melting process, the volume of fats increases and, therefore, the density decreases. The density of milk fat at 15 °C is between 0.935–0.943 and for coconut oil, 0.919–0.937 [57]. This may indicate that oil/fat used in research impacted products and dietary fiber used. This seems to be confirmed by the values between the individual concentrations. The density of PCS with AMF increased along with added fiber in product with BF, CF, PF (*p* < 0.05) and in sauces with OCO + CF and PF. However, the opposite relationship in the product with OCO + AF was noted. These results overlap with the results of hardness and the obtained products. For sauces with the addition of OCO, it can be seen that the increase in hardness is correlated with the density values. In products with AMF, the results were not so precise. We do not fully understand what causes these fluctuations in the density of the obtained products. Because they occurred mainly with sauces with the addition of AMF, we considered that animal fat used in production with the addition of specific fibers causes these changes, which may indicate the instability of the product structure.

Water activity (a_w_) and pH significantly impact fresh food, which offers a favorable environment for molds and yeast to grow [58]. The water activity of PCS with OCO lowered only for sauces with AF and BF; however, the decrease was not significant (*p* > 0.05). The lowest water activity was observed in the sample with 4% AF and AMF (0.979). In PCS with OCO, the change in fiber concentration influenced water activity only in the product with 4% BF (0.971) (*p* > 0.05). The a_w_ results between obtained processed cheese sauce are similar to each other. Coconut oil-in-water emulsion is described as an unstable solution [59], so it may be connected to dietary fibers added to products.

In turn, AMF is manufactured by removing the water phase from butter [60]. Among all tested sauces, PCS with bamboo fiber was characterized with the lowest value of a_w_, which is connected to its significant water holding capacity and fat binding capacities [61].

The FDA (Food and Drug Administration) recommends that food products with a pH above 4.6 is stable if its water activity is 0.85 or less. The PCS were obtained in pH 5.8, suggesting that observed a_w_ is the right condition for undesired microflora development. Therefore, it is critical to keep the obtained cheese sauces at the refrigeration temperature.

## 4. Conclusions

Rheological and textural properties of PCS were influenced by fat/oil and different amounts of dietary fibers. An increasing tendency of features including hardness, adhesiveness and viscosity in sauces with CF and PF addition prepared based on OCO or AMF was observed. OCO blends better in prepared, processed sauce and individual samples received higher viscosity values than those with anhydrous milk fat.

PCS presented shear-thinning behavior connected with yield stress and tan (δ) values, which indicated that all prepared samples had elastic properties.

The computer vision system (CVS) analysis showed that the prepared product’s color was associated with added fibers and different fat sources. The samples resulted in more intense colors, which may be visible to consumers (NBS). An increase in the density of sauces with CF and PF with AMF or OCO was correlated with their hardness. The water activity of processed cheese sauces obtained with OCO or AMF was not influenced by fiber type and concentration.

The observed textural, rheological and color features of the prepared sauces suggested that tested fibers can be used as an addition to processed cheese sauces based on acid casein and different oil/fat source. From a consumer point of view, sauces with the addition of AMF looked more like a commercial product, but both sources (OCO/AMF) seemed to be a good base according to physically and chemically stabilizing processed cheese sauces.

## Figures and Tables

**Figure 1 foods-10-00759-f001:**
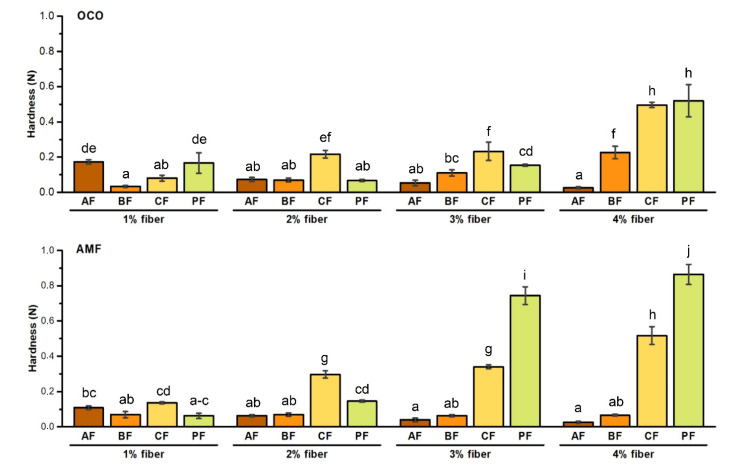
Effect of fiber type and concentration on hardness of PCS (processed cheese sauces) obtained on the basis of AC (acid casein) and WPC80 (whey protein concentrate) with OCO (organic coconut oil) or AMF (anhydrous milk fat). Letters (a–j) indicate significant differences at *p* < 0.05.

**Figure 2 foods-10-00759-f002:**
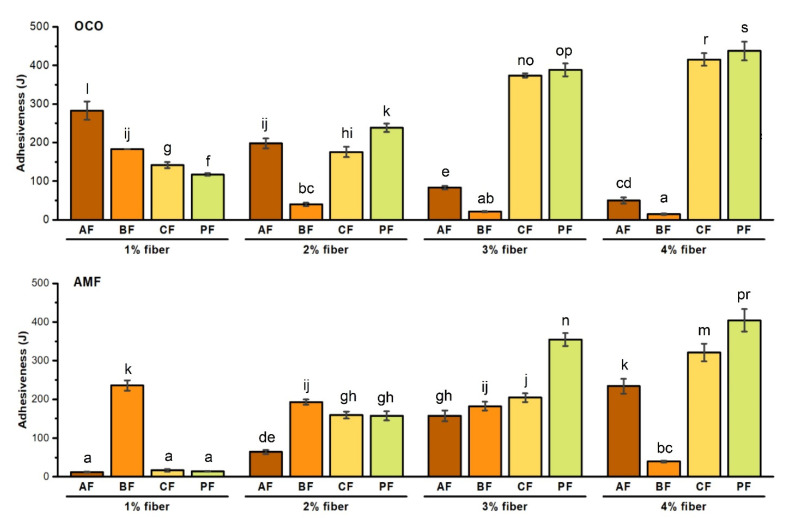
Effect of fiber type and concentration on adhesiveness of PCS (processed cheese sauces) obtained on the basis of AC (acid casein) and WPC80 (whey protein concentrate) with OCO (organic coconut oil) or AMF (anhydrous milk fat). Letters (a–p,r,s) indicate significant differences at *p* < 0.05.

**Figure 3 foods-10-00759-f003:**
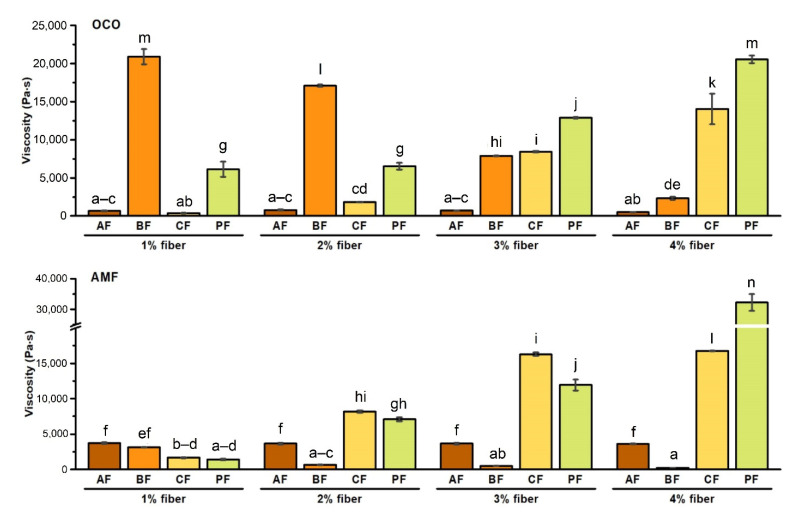
Effect of fiber type and concentration on viscosity of PCS (processed cheese sauces) obtained on the basis of AC (acid casein) and WPC80 (whey protein concentate) with OCO (organic coconut oil) or AMF (anhydrous milk fat) (*p* < 0.05). Letters (a–n) indicate significant differences at *p* < 0.05.

**Figure 4 foods-10-00759-f004:**
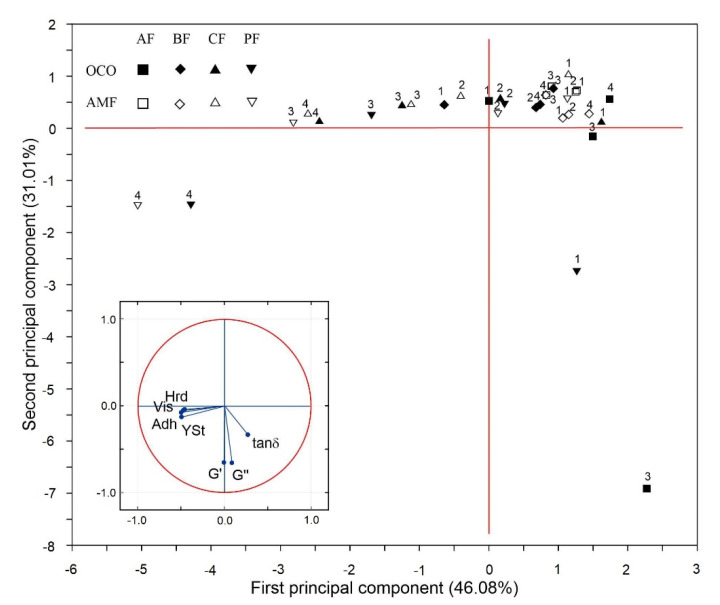
Principal component analysis (PCA) bi-plot for physical properties parameters (Hrd: hardness, Adh: adhesiveness, Vis: viscosity, YSt: yield stress, G′: storage modulus, G″: loss modulus, tan (δ): loss tangent) of acid casein PCS with different type of fat/oil (AFM or OCO), type of fiber (AF, BF, CF, PF) and fiber concentration (1%, 2%, 3%, 4%). Inset: correlation circle between variables all of the original variables included in a PCA in the first and second principal component factorial plane (correlation biplot).

**Figure 5 foods-10-00759-f005:**
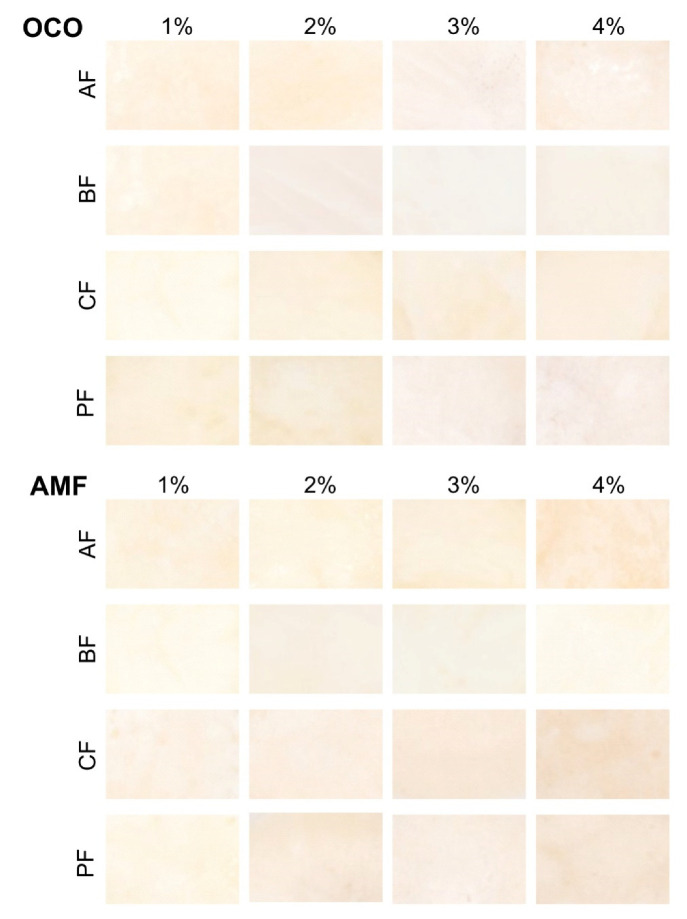
Effect of different fibers concentration of color of acid casein PCS (processed cheese sauces) with WPC80 (whey protein concentrate) and OCO (organic coconut oil) or AMF (anhydrous milk fat) measured with CVS (computer vision system).

**Table 1 foods-10-00759-t001:** Effects of the type of fat/oil, type of fiber and fiber concentration on physical parameters of acid casein processed cheese sauces (three-way ANOVA, F-ratio and *p*-values).

Dependent Variable	Item	Main Factor			Interactions			
		A: Fat	B: Fiber Type	C: Fiber Content	A × B	A × C	B × C	A × B × C
G′	F stat	683	1502	523	622	1281	825	1067
	*p*-value	<0.001	<0.001	<0.001	<0.001	<0.001	<0.001	<0.001
G″	F stat	788	1623	623	827	1212	801	1142
	*p*-value	<0.001	<0.001	<0.001	<0.001	<0.001	<0.001	<0.001
Tan (δ)	F stat	170	273	74	296	90	10	166
	*p*-value	<0.001	<0.001	<0.001	<0.001	<0.001	<0.001	<0.001
Yield stress	F stat	55,466	3592	30,609	837	13,302	929	2699
	*p*-value	<0.001	<0.001	<0.001	<0.001	<0.001	<0.001	<0.001
Hardness	F stat	529	92	330	105	149	39	46
	*p*-value	<0.001	<0.001	<0.001	<0.001	<0.001	<0.001	<0.001
Adhesiveness	F stat	744	95	421	270	444	128	113
	*p*-value	<0.001	<0.001	<0.001	<0.001	<0.001	<0.001	<0.001
Viscosity	F stat	919	9	429	647	572	161	69
	*p*-value	<0.001	0.004	<0.001	<0.001	<0.001	<0.001	<0.001
L*	F stat	241	65	59	119	7	36	14
	*p*-value	<0.001	<0.001	<0.001	<0.001	<0.001	<0.001	<0.001
a*	F stat	144	96	17	95	17	7	20
	*p*-value	<0.001	<0.001	<0.001	<0.001	<0.001	<0.001	<0.001
b*	F stat	240	388	31	270	32	71	51
	*p*-value	<0.001	<0.001	<0.001	<0.001	<0.001	<0.001	<0.001
Chroma	F stat	251	381	22	252	29	65	47
	*p*-value	<0.001	<0.001	<0.001	<0.001	<0.001	<0.001	<0.001
Hue	F stat	18	36	78	115	22	39	24
	*p*-value	<0.001	<0.001	<0.001	<0.001	<0.001	<0.001	<0.001
Density	F stat	173	75	161	110	24	88	15
	*p*-value	<0.001	<0.001	<0.001	<0.001	<0.001	<0.001	<0.001
Water activity	F stat	5	0	2	2	2	0	2
	*p*-value	0.003	0.834	0.075	0.185	0.040	0.829	0.060

**Table 2 foods-10-00759-t002:** Effect of different fibers concentration on PCS (processed cheese sauces) obtained on the basis of AC (acid casein) with OCO (organic coconut oil) or AC (acid casein) with AMF (anhydrous milk fat) on value of G′ and G″ moduli, tan (δ) and yield stress.

Fat/Oil	Fiber Type	G′ (Pa)				G″ (Pa)				tan (δ)				Yield Stress (Pa)			
		Fiber Content (%)				Fiber Content (%)				Fiber Content (%)				Fiber Content (%)			
		1	2	3	4	1	2	3	4	1	2	3	4	1	2	3	4
OCO	AF	3.25 ^a^ ±0.69	0.70 ^a^ ±0.12	60,086 ^m^ ±8319	785 ^cd^ ±79	1.17 ^a^ ±0.55	0.55 ^a^ ±0.09	43,014 ^i^ ±5590	361 ^b–d^ ±74	0.40 ^j^ ±0.22	0.79 ^r^ ±0.10	0.72 ^p^ ±0.08	0.47 ^l^ ±0.12	601 ^k^ ±1	206 ^f^ ±1	212 ^f^ ±1	1.04 ^a^ ±0.03
	BF	13,384 ^j^ ±35	568 ^b–d^ ±7	383 ^bc^ ±1	3774 ^hi^ ±10	970 ^de^ ±1	278 ^a–c^ ±28	134 ^a–c^ ±1	1235 ^ef^ ±25	0.07 ^a^ ±0.01	0.49 ^lm^ ±0.05	0.35 ^g–i^ ±0.01	0.33 ^d–g^ ±0.01	1.82 ^a^ ±0.01	40.3 ^b^ ±0.2	81.5 ^c^ ±1.6	407 ^h^ ±2
	CF	11.4 ^a^ ±0.1	2703 ^e–h^ ±211	546 ^b–d^ ±22	3122 ^f–h^ ±22	7.68 ^a^ ±0.14	775 ^c–e^ ±66	181 ^a–c^ ±32	950 ^de^ ±5	0.67 ^o^ ±0.01	0.29 ^cd^ ±0.05	0.33 ^e–g^ ±0.07	0.30 ^de^ ±0.01	47.9 ^b^ ±0.1	415 ^hi^ ±1	894 ^m^ ±12	963 ^n^ ±12
	PF	23,837 ^l^ ±841	951 ^cd^ ±13	2824 ^e–h^ ±38	13,784 ^j^ ±210	16,727 ^h^ ±1364	394 ^b–d^ ±43	874 ^c–e^ ±12	5413 ^g^ ±545	0.70 ^op^ ±0.07	0.41 ^jk^ ±0.05	0.31 ^d–f^ ±0.01	0.39 ^ih^ ±0.04	205 ^f^ ±1	415 ^hi^ ±3	1254 ^r^ ±18	2973 ^t^ ±46
AMF	AF	1158 ^c–e^ ±36	844 ^cd^ ±10	633 ^b–d^ ±4	177 ^b^ ±3	411 ^b–e^ ±6	322 ^b–d^ ±11	210 ^a–c^ ±2	74 ^ab^ ±7	0.35 ^g–i^ ±0.01	0.38 ^h–j^ ±0.01	0.33 ^d–g^ ±0.01	0.42 ^k^ ±0.04	17.3 ^a^ ±0.3	16.1 ^a^ ±0.4	6.42 ^a^ ±0.4	0.66 ^a^ ±0.36
	BF	6.12 ^a^ ±0.09	1744 ^d–e^ ±13	2650 ^e–h^ ±30	4717 ^i^ ±117	3.82 ^a^ ±0.12	906 ^c–e^ ±11	944 ^de^ ±8	1922 ^f^ ±217	0.62 ^n^ ±0.01	0.52 ^m^ ±0.01	0.32 ^d–g^ ±0.01	0.41 ^jk^ ±0.05	108 ^d^ ±3	161 ^e^ ±3	219 ^f^ ±1	6.81 ^a^ ±0.01
	CF	29.9 ^ab^ ±0.02	3451 ^gh^ ±77	336 ^bc^ ±11	2022 ^d–f^ ±10	7.50 ^a^ ±0.05	769 ^c–e^ ±15	116 ^a–c^ ±2	523 ^b–e^ ±4	0.25 ^bc^ ±0.01	0.22 ^b^ ±0.01	0.34 ^e–h^ ±0.01	0.26 ^bc^ ±0.01	6.81 ^a^ ±0.01	338 ^g^ ±2	562 ^j^ ±5	1217 ^p^ ±13
	PF	3773 ^hi^ ±10	1269 ^c–e^ ±57	2511 ^e–g^ ±16	19,815 ^k^ ±3069	1177 ^e^ ±38	573 ^b–e^ ±15	770 ^c–e^ ±4	5242 ^g^ ±1756	0.31 ^d–f^ ±0.01	0.45 ^kl^ ±0.02	0.31 ^d–f^ ±0.01	0.26 ^bc^ ±0.05	429 ^i^ ±2	706 ^l^ ±5	1143 ^o^ ±4	1835 ^s^ ±1

Data are presented as means ± SD (standard deviation). ^a–t^ Means within columns and rows for each parameter (G′, G″, tan (δ) or yield stress) with different superscripts are significantly different (*p* < 0.05, Duncan’s test).

**Table 3 foods-10-00759-t003:** Effect of different fibers concentration color (CIE Lab standards) of acid casein PCS (processed cheese sauces) with WPC80 (whey protein concentrate) and OCO (organic coconut oil) or AMF (anhydrous milk fat).

Fat/ Oil	Fiber Type	L*				a*				b*				Chroma				Hue			
		Fiber Content (%)				Fiber Content (%)				Fiber Content (%)				Fiber Content (%)				Fiber Content (%)			
		1	2	3	4	1	2	3	4	1	2	3	4	1	2	3	4	1	2	3	4
OCO	AF	85.0 ^gh^ ±1.5	85.9 ^h–k^ ±0.9	86.9 ^k–n^ ±0.4	85.0 ^gh^ ±0.6	6.14 ^n^ ±0.69	5.71 ^m^ ±0.49	4.29 ^f–i^ ±0.49	5.57 ^lm^ ±0.53	19.4 ^hi^ ±2.2	19.9 ^h–j^ ±1.6	7.86 ^a^ ±1.21	14.7 ^cd^ ±1.5	20.4 ^ij^ ±2.3	20.7 ^ij^ ±1.6	8.96 ^ab^ ±1.23	15.7 ^d–f^ ±1.5	17.6 ^k–n^ ±1.6	16.1 ^i–m^ ±1.1	28.8 ^r^ ±3.0	20.8 ^op^ ±2.1
	BF	88.9 ^p^ ±0.4	88.6 ^op^ ±0.5	87.1 ^l–n^ ±1.5	87.1 ^l–n^ ±0.9	2.43 ^a–d^ ±0.53	2.43 ^a–d^ ±0.53	2.14 ^a–c^ ±0.69	3.71 ^e–g^ ±0.49	9.57 ^b^ ±0.53	9.29 ^b^ ±0.76	7.14 ^a^ ±2.11	9.29 ^b^ ±0.95	9.88 ^b^ ±0.56	9.61 ^b^ ±0.81	7.47 ^a^ ±2.19	10.0 ^b^ ±0.9	14.2 ^g–j^ ±3.0	14.6 ^h–k^ ±2.7	16.8 ^j–n^ ±3.3	21.9 ^p^ ±3.1
	CF	88.1 ^n–p^ ±0.7	87.6 ^m–p^ ±0.5	86.6 ^j–m^ ±0.5	86.4 ^i–l^ ±0.8	1.71 ^a^ ±0.76	2.00 ^a–c^ ±0.82	1.85 ^ab^ ±0.69	2.71 ^cd^ ±0.76	8.57 ^ab^ ±0.53	9.00 ^ab^ ±0.58	8.14 ^ab^ ±1.99	9.71 ^b^ ±0.76	8.77 ^ab^ ±0.57	9.25 ^ab^ 0.59	8.39 ^ab^ ±1.99	10.1 ^b^ ±0.7	11.3 ^e–g^ ±4.7	12.5 ^f–h^ ±5.0	13.5 ^g–i^ ±5.8	15.7 ^i–l^ ±4.8
	PF	81.1 ^bc^ ±1.5	80.0 ^ab^ ±0.1	79.6 ^a^ ±2.5	83.0 ^de^ ±1.2	3.00 ^de^ ±0.58	2.71 ^cd^ ±0.76	5.43 ^k–m^ ±1.27	4.57 ^h–j^ ±0.53	26.3 ^l^ ±1.5	26.9 ^l^ ±1.9	24.1 ^k^ ±2.9	8.00 ^ab^ ±0.82	26.5 ^kl^ ±1.5	27.0 ^l^ ±1.9	24.7 ^k^ ±3.0	9.22 ^ab^ ±0.93	6.50 ^ab^ ±1.19	5.71 ^a^ ±1.23	12.5 ^f–h^ ±1.6	29.7 ^r^ ±1.9
AMF	AF	85.6 ^g–j^ ±1.0	85.0 ^gh^ ±1.6	82.6 ^de^ ±1.5	82.0 ^cd^ ±0.6	3.57 ^ef^ ±0.79	3.71 ^e–g^ ±0.76	4.43 ^g–i^ ±0.79	7.86 ^o^ ±0.38	20.3 ^ij^ ±2.5	21.6 ^j^ ±1.9	26.9 ^l^ ±1.6	23.7 ^k^ ±1.6	20.6 ^ij^ ±2.6	21.9 ^j^ ±1.9	27.2 ^l^ ±1.6	25.0 ^k^ ±1.6	9.90 ^c–f^ ±1.50	9.77 ^c–f^ ±1.84	9.37 ^b–e^ ±1.68	18.4 ^l–o^ ±0.8
	BF	88.1 ^n–p^ ±1.6	86.9 ^k–n^ ±0.9	87.3 ^m–o^ ±0.8	87.9 ^m–p^ ±1.2	2.57 ^b–d^ ±0.53	4.14 ^f–h^ ±0.38	2.00 ^a–c^ ±0.01	2.14 ^a–c^ ±0.38	21.3 ^j^ ±4.1	14.7 ^cd^ ±1.7	16.6 ^d–g^ ±1.0	16.9 ^fg^ ±1.7	21.4 ^j^ ±4.1	15.3 ^c–e^ ±1.7	16.7 ^e–g^ ±1.0	17.0 ^e–g^ ±1.7	7.00 ^a–c^ ±1.67	15.8 ^i–l^ ±1.4	6.90 ^a–c^ ±0.41	7.32 ^a–d^ ±1.59
	CF	85.3 ^g–i^ 0.5	83.4 ^ef^ 1.0	80.3 ^ab^ 0.5	79.7 ^a^ 0.8	4.71 ^h–k^ ±0.49	5.29 ^j–l^ ±0.76	5.71 ^jk^ ±0.49	5.57 ^lm^ ±0.79	14.9 ^c–e^ ±1.1	16.0 ^d–f^ ±1.1	17.7 ^fg^ ±1.1	18.1 ^gh^ ±0.7	15.6 ^c–e^ ±1.1	16.9 ^e–g^ ±1.5	17.7 ^f–h^ ±1.1	19.0 ^hi^ ±0.6	17.6 ^k–n^ ±1.5	18.2 ^l–o^ ±1.6	18.9 ^m–o^ ±1.5	17.1 ^j–n^ ±2.6
	PF	88.0 ^m–p^ ±1.4	84.3 ^fg^ ±1.1	81.9 ^cd^ ±1.3	80.0 ^ab^ ±0.8	3.14 ^de^ ±0.38	4.43 ^g–i^ ±0.53	5.00 ^i–l^ ±0.82	5.57 ^lm^ ±0.53	17.6 ^fg^ ±1.0	13.0 ^c^ ±1.3	13.1 ^c^ ±1.6	15.7 ^d–f^ ±1.0	17.9 ^gh^ ±1.0	13.7 ^c^ ±1.3	14.1 ^cd^ ±1.7	16.7 ^e–g^ ±1.0	10.1 ^d–f^ ±0.9	18.9 ^l–o^ ±1.9	20.8 ^op^ ±1.5	19.5 ^n–p^ ±1.2

Data are presented as means ± SD (standard deviation). ^a–r^ Means within columns and rows for each parameter (L*, a*, b*, Chroma or Hue) with different superscripts are significantly different (*p* < 0.05, Duncan’s test)

**Table 4 foods-10-00759-t004:** Effect of different fibers, AC (acid casein), WPC80 (whey protein concentrate) and OCO (organic coconut oil) or AMF (anhydrous milk fat) on color differences of PCS (processed cheese sauces) expressed by NBS units (∆E × 0.92) measured with CVS (computer vision system).

OCO																			AMF															
			AF(%)				BF(%)				CF(%)				PF(%)				AF(%)				BF(%)				CF(%)				PF(%)			
			1	2	3	4	1	2	3	4	1	2	3	4	1	2	3	4	1	2	3	4	1	2	3	4	1	2	3	4	1	2	3	4
OCO	AF(%)	1	-																															
		2	1.03	-																														
		3	10.9	11.2	-																													
		4	4.3	4.9	6.6	-																												
	BF(%)	1	10.3	10.3	3.0	6.6	-																											
		2	10.4	10.5	2.7	6.6	0.4	-																										
		3	12.0	12.2	2.09	7.9	2.8	2.4	-																									
		4	9.8	10.0	1.4	5.6	2.05	1.8	2.4	-																								
	CF(%)	1	10.8	11.2	2.7	7.2	1.4	1.04	1.7	2.16	-																							
		2	10.6	10.7	2.4	6.6	2.4	1.04	1.8	1.7	0.7	-																						
		3	11.2	11.4	2.3	7.09	2.6	2.2	1.06	2.06	1.4	1.2	-																					
		4	9.5	9.8	2.3	5.4	2.4	2.08	2.5	1.19	2.1	1.4	1.7	-																				
	PF(%)	1	7.8	7.8	17.8	11.5	17.0	17.1	18.5	16.6	17.6	17.0	17.5	16.0	-																			
		2	8.9	8.9	18.7	12.4	17.9	18.0	19.3	17.5	18.5	17.9	18.3	16.9	1.2	-																		
		3	6.6	7.0	16.4	10.0	16.1	16.2	17.3	15.4	16.6	16.0	16.4	14.9	3.3	3.6	-																	
		4	10.7	11.3	3.6	6.5	5.9	5.6	4.5	4.03	5.4	4.9	4.2	3.9	17.0	17.7	15.2	-																
AMF	AF(%)	1	2.6	2.02	11.5	5.5	10.4	10.6	12.3	10.2	11.2	10.7	11.3	9.8	6.9	8.0	6.8	11.6	-															
		2	3.02	2.6	12.8	6.6	11.7	11.9	13.5	11.5	12.5	11.9	12.6	11.1	5.7	6.8	5.7	12.7	1.3	-														
		3	7.4	7.2	17.9	11.5	17.1	17.2	18.8	16.7	17.8	17.2	17.8	16.3	2.0	2.9	3.9	17.4	6.7	5.4	-													
		4	5.08	5.4	15.6	9.0	15.3	15.4	16.8	15.6	16.0	15.4	15.9	14.3	5.1	5.9	3.2	14.8	6.0	5.1	4.4	-												
	BF(%)	1	4.7	3.8	12.5	7.3	10.8	11.1	13.1	11.1	11.7	11.3	12.2	10.8	7.9	9.06	8.6	13.2	2.6	3.1	7.4	7.8	-											
		2	5.01	5.08	6.3	2.2	5.3	5.4	7.2	5.0	6.2	5.6	6.4	4.8	12.0	13.0	11.0	7.1	5.3	6.6	11.3	10.0	6.3	-										
		3	5.06	4.7	8.3	4.3	6.6	6.8	8.7	6.9	7.4	7.0	7.8	6.4	10.6	11.6	10.4	9.2	4.0	5.3	10.7	9.8	4.4	2.7	-									
		4	5.09	4.7	8.6	4.6	6.8	7.04	9.0	7.2	7.7	7.3	8.2	6.8	10.7	11.7	10.6	9.6	4.0	5.3	10.6	9.8	4.1	2.9	0.6	-								
	CF(%)	1	4.4	4.7	6.7	0.9	6.3	6.3	7.7	5.5	6.9	6.3	6.9	5.2	11.3	12.2	10.0	6.7	5.1	6.2	11.3	9.1	6.7	1.6	3.5	3.8	-							
		2	3.0	3.8	8.8	2.4	8.7	8.7	9.9	7.8	9.2	8.6	9.0	7.4	9.3	10.2	7.7	8.0	4.2	5.0	9.4	7.0	6.5	3.9	4.7	5.1	2.5	-						
		3	4.6	5.5	11.0	5.1	11.3	11.3	12.0	10.1	11.6	11.0	11.1	9.7	8.3	8.9	5.9	9.3	5.8	6.1	8.8	6.1	8.4	6.8	7.4	7.8	5.4	3.0	-					
		4	5.05	5.9	11.6	5.8	11.9	11.9	12.6	10.7	12.2	11.6	11.7	10.2	8.01	8.5	5.5	9.8	6.1	6.1	8.6	6.0	8.7	7.4	7.8	8.3	6.0	3.6	0.7	-				
	PF(%)	1	4.2	3.7	9.08	4.4	7.5	7.7	9.7	7.7	8.4	8.0	8.9	7.4	10.2	11.3	10.0	10.0	3.4	4.6	10.0	9.0	3.4	3.0	1.5	1.1	3.8	4.7	7.5	8.0	-			
		2	6.1	6.6	5.3	2.0	5.6	5.5	6.3	4.3	5.9	5.3	5.5	3.9	12.7	13.5	11.1	4.8	6.9	8.0	13.7	10.6	8.6	2.9	4.9	5.3	2.0	3.6	5.8	6.4	5.6	-		
		3	6.5	7.3	6.7	3.3	7.6	7.5	7.7	6.05	7.7	7.0	6.9	5.6	12.4	13.0	10.3	4.8	7.6	8.4	12.7	10.1	9.7	4.9	6.5	7.0	3.5	3.6	4.5	5.1	7.2	2.3	-	
		4	5.7	6.7	9.7	4.7	10.4	10.3	10.7	9.0	10.5	9.9	9.8	8.5	10.2	10.6	7.7	7.7	6.9	7.3	10.6	7.9	9.5	6.5	7.5	8.0	5.0	3.3	1.9	2.2	7.9	4.8	3	-

**Table 5 foods-10-00759-t005:** Effect of different fibers concentration on density and water activity of acid casein PCS (processed cheese sauces) with WPC80 (whey protein concentrate) and OCO (organic coconut oil) or AMF (anhydrous milk fat).

Fat/Oil	Fiber Type	Density (g/mL)				Water Activity			
		Fiber Content (%)				Fiber Content (%)			
		1	2	3	4	1	2	3	4
OCO	AF	1.071 ^g–i^ ±0.001	1.067 ^f–h^ ±0.002	1.065 ^e–g^ ±0.004	1.061 ^c–e^ ±0.001	0.998 ^c^ ±0.001	0.989 ^bc^ ±0.002	0.992 ^bc^ ±0.003	0.979 ^b^ ±0.001
	BF	1.046 ^a^ ±0.001	1.047 ^a^ ±0.004	1.058 ^bc^ ±0.002	1.054 ^b^ ±0.003	0.990 ^bc^ ±0.001	0.980 ^bc^ ±0.001	0.981 ^bc^ ±0.002	0.981 ^bc^ ±0.001
	CF	1.064 ^d–f^ ±0.002	1.068 ^f–h^ ±0.002	1.068 ^f–h^ ±0.00	1.070 ^g–i^ ±1.002	0.985 ^bc^ ±0.004	0.988 ^bc^ ±0.003	0.988 ^bc^ ±0.001	0.990 ^bc^ ±0.001
	PF	1.072 ^hi^ ±0.002	1.071 ^g–i^ ±0.001	1.081 ^j–l^ ±0.007	1.091 ^n^ ±0.007	0.984 ^bc^ ±0.001	0.985 ^bc^ 0.001	0.980 ^bc^ ±0.001	0.990 ^bc^ ±0.001
AMF	AF	1.045 ^a^ ±0.004	1.06 ^cd^ ±0.003	1.044 ^a^ ±0.002	1.078 ^jk^ ±0.004	0.988 ^bc^ ±0.001	0.985 ^bc^ 0.003	0.986 ^bc^ ±0.001	0.984 ^bc^ ±0.001
	BF	1.068 ^f–h^ ±0.001	1.068 ^f–h^ ±0.001	1.076 ^ij^ ±0.005	1.086 ^l–n^ ±0.005	0.988 ^bc^ ±0.001	0.984 ^bc^ 0.001	0.985 ^bc^ ±0.001	0.971 ^a^ ±0.005
	CF	1.045 ^a^ ±0.005	1.084 ^k–m^ ±0.005	1.080 ^jk^ ±0.003	1.090 ^n^ ±0.006	0.996 ^bc^ ±0.004	0.990 ^bc^ 0.003	0.990 ^bc^ ±0.001	0.980 ^bc^ ±0.001
	PF	1.056 ^bc^ ±0.002	1.079 ^jk^ ±0.001	1.088 ^mn^ ±0.001	1.098 ^o^ ±0.001	0.980 ^bc^ ±0.001	0.990 ^bc^ 0.001	0.990 ^bc^ ±0.001	0.990 ^bc^ ±0.003

Data are presented as means ± SD (standard deviation). ^a–o^ Means within columns and rows for each parameter (Density or Water activity) with different superscripts are significantly different (*p* < 0.05, Duncan’s test).

## Data Availability

Data reported in this manuscript will be available upon request.

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
