# Peer review of "The Influence of Dietary Fibers on Physicochemical Properties of Acid Casein Processed Cheese Sauces Obtained with Whey Proteins and Coconut Oil or Anhydrous Milk Fat"

_foods, 2021, doi:10.3390/foods10040759_

Round 1

Reviewer 1 Report

To the authors

The paper is interesting and overall well written. There are only a few points to be clarified in Materials and Methods and some minor remarks. In particular, the number of observations for each preparation should be specified. In particular, compliments for the Results and Discussion section, that is very complete and interesting.

I would suggest a minor revision.

Specific comments:

Abstract, last line: in the abstract the acronyms OCO and AMF are not defined: they should be defined or avoided.

Materials and methods: the experimental plan is not fully clear: in particular it is not specified how many samples have been prepared and analysed for each type of preparation; i.e., the number of observations at the beginning of Materials and Methods should be specified.

Page 3 Line 121-122: it is better to write “was described” instead of ”was describing”.

Page 3 Line 123: too much space between “blue)” and “using.

Page 3 lines 143-154: the statistical program used is not specified; should be added.

Page 3 Table 1: the word “Adhesiveness” should stay in the same line in the table.

Page 6 Line 226: the reference “(Newton & Gortner, 1922)” should be cited with a progressive number, and within a square parenthesis, as requested by the rules of the Journal.

Make attention to the number of page, on the top of the pages: they are not correct, because, after the tables, sometimes the number of page is not progressive.

Figure 5: the figure is not so clear: make sure it is printed with a very high resolution, because otherwise you will not be able to perceive the differences between the various panels.

Figure 5 should be cited also in the text.

Page 2 after figure 5, Line 462: I think that Table S1 is interesting and I would not put it in Supplementary Material but as a normal figure of the paper. Generally I am not very much in favour of Supplementary Material, I think only tables with statistical parameters should be put there, not tables with results.

Page 3 after figure 5, Line 503: subscript the w of aw.

Author Response

University of Life Sciences in Lublin

Faculty of Food Sciences and Biotechnology

Department of Milk Technology and Hydrocolloids

Skromna 8, 20-704 Lublin

Poland

Phone: +48 81 4623350

Fax:     +48 81 4623345

E-mail: bartosz.solowiej@up.lublin.pl

March 25, 2021,

Dear Reviewer,

Our manuscript entitled “The influence of dietary fibers on physicochemical properties of acid casein processed cheese sauces obtained with whey proteins and coconut oil or anhydrous milk fat” (Ref. No.: foods-1149884) has been revised and is being re-submitted for publication in Foods – Special Issue “New Frontiers in Dairy Technology and Hydrocolloids”

We have carefully considered each of the comments and made the appropriate revisions in the manuscript. An itemized list of our responses to each of the comments is included below.

Thank you for your kind attention.

Yours faithfully,

Bartosz Sołowiej

We have corrected our manuscript with regard to Reviewers’ comments:

Reviewer #1:  green color

Reviewer #2: blue color

Reviewer #3: yellow color

If Reviewers had the same comments – underline

Reviewer #1: Review of Manuscript Number: foods-1149884

Title: The influence of dietary fibers on physicochemical properties of acid casein processed cheese sauces obtained with whey proteins and coconut oil or anhydrous milk fat

To the authors

The paper is interesting and overall well written. There are only a few points to be clarified in Materials and Methods and some minor remarks. In particular, the number of observations for each preparation should be specified. In particular, compliments for the Results and Discussion section, that is very complete and interesting.

I would suggest a minor revision.

Thank you very much for your opinion about the manuscript.

  1. Specific comments:

Abstract, last line: in the abstract the acronyms OCO and AMF are not defined: they should be defined or avoided.

 Thank you for your comment. We have added information about used acronyms: organic coconut oil (OCO) and anhydrous milk fat (AMF) (lines 22-23).

  1. Materials and methods: the experimental plan is not fully clear: in particular it is not specified how many samples have been prepared and analysed for each type of preparation; i.e., the number of observations at the beginning of Materials and Methods should be specified.

Thank you for your comment. We have specified information about how many samples have been prepared and analyzed for each type of preparation. At the beginning of Material and Methods (lines 120-123) and also to clarify mentioned section we also have added in lines 142-144 (Penetration test), 149 (Viscosity), 156 (Viscoelastic Properties), 159 (Colorimetric Measurements), 173-174 (Water activity), 178-179 (Density) additional information about the number of tested samples.

  1. Page 3 Line 121-122: it is better to write “was described” instead of ”was describing”.

Thank you for your comment. We have corrected the mentioned fragment in the text (line 160).

  1. Page 3 Line 123: too much space between “blue)” and “using.

Thank you for your comment. We have corrected the mentioned fragment in the text (lines 161).

  1. Page 3 lines 143-154: the statistical program used is not specified; should be added.

Thank you for your comment. In lines 192-193 (All calculations were carried out with Statistica software (v. 13.3, TIBCO Software Inc., Palo Alto, CA, USA) we have mentioned an information about software that was used.

  1. Page 3 Table 1: the word “Adhesiveness” should stay in the same line in the table.

Thank you for your comment. We have corrected the description in Table 1.

  1. Page 6 Line 226: the reference “(Newton & Gortner, 1922)” should be cited with a progressive number, and within a square parenthesis, as requested by the rules of the Journal.

Thank you for your comment. We have clarified cited information about acacia gum and we decided to use only one reference to describe the mentioned matter - Sanchez et al. (line 263), which is the newest one, with progressive number 33.

  1. Make attention to the number of page, on the top of the pages: they are not correct, because, after the tables, sometimes the number of page is not progressive.

Thank you for youromment. It was established automatically. We will contact the Editor's Office to c correct this mistake.

  1. Figure 5: the figure is not so clear: make sure it is printed with a very high resolution, because otherwise you will not be able to perceive the differences between the various panels.

Thank you for your comment. We will contact the Editor's office to make sure it is printed with a very high resolution to perceive the differences between the various panels.

  1. Figure 5 should be cited also in the text.

Thank you for your comment. In line 502 we have mentioned the Figure 5 in the text.

  1. Page 2 after figure 5, Line 462: I think that Table S1 is interesting and I would not put it in Supplementary Material but as a normal figure of the paper. Generally I am not very much in favour of Supplementary Material, I think only tables with statistical parameters should be put there, not tables with results.

Thank you for your comment. We have tried to put the Table S1 in the main text, but at its current size is too large to fit in the text. We will ask the Editor’s Office to help us to place the Table S1 in the text.

  1. Page 3 after figure 5, Line 503: subscript the w of aw.

Thank you for your comment. We have corrected the aw (line 503 before) - line 550 now.

Reviewer 2 Report

Dear Editor and Authors,

I send you my review about the paper “The influence of dietary fibers on physicochemical properties of acid casein processed cheese sauces obtained with whey proteins and coconut oil or anhydrous milk fat”.

The purpose of the paper, as reported in the aim was to test the effect of different dietary fibers on texture, rheological properties, color, density, and processed cheese sauces' water activity based on acid casein, WPC80, and anhydrous milk fat or organic coconut oil.

The paper result well written and structured, however, in this form it show some lacks.

The introduction, result well written and adequately supports the aim of the paper.

However, in the introduction should be better explained the originality of this paper, that in this version, result a little lacking. To improve the originality of the paper, I suggest to stress, in the introduction, the differences of the caracteristics between the sauce object of this paper and the ones of other similary products.

The experimental design is well structured and adequate to the aim. Moreover the analytical methods result adequate to the experimental design and, also, adequately supported by references. However, in the section “Materials” should be reported the number of trial performed.

The results is well presented and adequately discussed, also, in relation to the references reported that it is complete.

Data in the table is well shown, but, however, I suggest to the Authors to paginate the table 1, that in my version result split between pages 4 and 5. Furthermore, also the figure 5 should be paginate in a single page.

Moreover, always the figure 5, is both not mentioned and it is not discussed in the in the text. Figure 5 should be discussed in the text or delete.

Finally, the conclusions are in accordance with the data discussed and with the aim of the document, however, the text from line 515 to line 530 is a repetition of the comment of the discussion, this part of the text should be rewritten.

Best regards

Author Response

University of Life Sciences in Lublin

Faculty of Food Sciences and Biotechnology

Department of Milk Technology and Hydrocolloids

Skromna 8, 20-704 Lublin

Poland

Phone: +48 81 4623350

Fax:     +48 81 4623345

E-mail: bartosz.solowiej@up.lublin.pl

March 25, 2021,

Dear Reviewer,

Our manuscript entitled “The influence of dietary fibers on physicochemical properties of acid casein processed cheese sauces obtained with whey proteins and coconut oil or anhydrous milk fat” (Ref. No.: foods-1149884) has been revised and is being re-submitted for publication in Foods – Special Issue “New Frontiers in Dairy Technology and Hydrocolloids”

We have carefully considered each of the comments and made the appropriate revisions in the manuscript. An itemized list of our responses to each of the comments is included below.

Thank you for your kind attention.

Yours faithfully,

Bartosz Sołowiej

We have corrected our manuscript with regard to Reviewers’ comments:

Reviewer #1:  green color

Reviewer #2: blue color

Reviewer #3: yellow color

If Reviewers had the same comments – underline

Reviewer #2: Review of Manuscript Number: foods-1149884

Title: The influence of dietary fibers on physicochemical properties of acid casein processed cheese sauces obtained with whey proteins and coconut oil or anhydrous milk fat

Dear Editor and Authors,

I send you my review about the paper “The influence of dietary fibers on physicochemical properties of acid casein processed cheese sauces obtained with whey proteins and coconut oil or anhydrous milk fat”.

The purpose of the paper, as reported in the aim was to test the effect of different dietary fibers on texture, rheological properties, color, density, and processed cheese sauces' water activity based on acid casein, WPC80, and anhydrous milk fat or organic coconut oil.

The paper result well written and structured, however, in this form it show some lacks.

Thank you very much for your opinion about the manuscript.

  1. The introduction, result well written and adequately supports the aim of the paper.

However, in the introduction should be better explained the originality of this paper, that in this version, result a little lacking. To improve the originality of the paper, I suggest to stress, in the introduction, the differences of the characteristics between the sauce object of this paper and the ones of other similar products.

Thank you for your comment. In lines 44-69 we have better explained the originality of our research. According to your suggestion, we have better explained the differences between the processed cheese sauces formula presented in this paper and other sauces/commercial products.

References added to the Introduction section:

  1. SzafraÅ„ska, J.O.; SoÅ‚owiej, B.G. Cheese sauces: Characteristics of ingredients, manufacturing methods, microbiological and sensory aspects. J. Food Process Eng. 2020, 43, 1–14, doi:10.1111/jfpe.13364.
  2. Shinde, G.; Kumar, R.; Chauhan, S.K.; Shinde, G.; Subramanian, V.; Nadanasabapathi, S. Whey Proteins: A potential ingredient for food industry- A review. Asian J. Dairy Food Res. 2018, doi:10.18805/ajdfr.dr-1389.
  3. Sarode A.R., Sawale P.D., Khedkar C.D., Kalyankar S.D., P.R.D. Casein and caseinates. Methods of Manufacture. In The Encyclopedia of Food and Health; Caballero, B., Finglas, P., Toldrá, F., Ed.; Oxford: Academic Press, 2016; pp. 676–682.
  4. Wang, J.; Wang, X.; Li, J.; Chen, Y.; Yang, W.; Zhang, L. Effects of dietary coconut oil as a medium-chain fatty acid source on performance, carcass composition and serum lipids in male broilers. Asian-Australasian J. Anim. Sci. 2015, 28, 223–230, doi:10.5713/ajas.14.0328.
  5. Deen, A.; Visvanathan, R.; Wickramarachchi, D.; Marikkar, N.; Nammi, S.; Jayawardana, B.C.; Liyanage, R. Chemical composition and health benefits of coconut oil: an overview. J. Sci. Food Agric. 2020, 101, 2182–2193, doi:10.1002/jsfa.10870.
  6. Herrera-Meza, M.S.; Mendoza-López, M.R.; García-Barradas, O.; Sanchez-Otero, M.G.; Silva-Hernández, E.R.; Angulo, J.O.; Oliart-Ros, R.M. Dietary anhydrous milk fat naturally enriched with conjugated linoleic acid and vaccenic acid modify cardiovascular risk biomarkers in spontaneously hypertensive rats. Int. J. Food Sci. Nutr. 2013, 64, 575–586, doi:10.3109/09637486.2013.763908.
  7. Sołowiej, B.G.; Nastaj, M.; Szafrańska, J.O.; Muszyński, S.; Gustaw, W.; Tomczyńska-Mleko, M.; Mleko, S. Effect of emulsifying salts replacement with polymerised whey protein isolate on textural, rheological and melting properties of acid casein model processed cheeses. Int. Dairy J. 2020, 105, doi:10.1016/j.idairyj.2020.104694.
  8. Barber, T.M.; Kabisch, S.; Pfeiffer, A.F.H.; Weickert, M.O. The health benefits of dietary fibre. Nutrients 2020, 12, 1–17, doi:10.3390/nu12103209.
  1. The experimental design is well structured and adequate to the aim. Moreover the analytical methods result adequate to the experimental design and, also, adequately supported by references. However, in the section “Materials” should be reported the number of trial performed.

Thank you for your comment. We have specified information about how many samples have been prepared and analyzed for each type of preparation. At the beginning of Material and Methods (lines 120-123) and also to clarify mentioned section we also have added in lines 142-144 (Penetration test), 149 (Viscosity), 156 (Viscoelastic Properties), 159 (Colorimetric Measurements), 173-174 (Water activity), 178-179 (Density) additional information about the number of tested samples.

  1. The results is well presented and adequately discussed, also, in relation to the references reported that it is complete.

Thank you for your comment.

  1. Data in the table is well shown, but, however, I suggest to the Authors to paginate the table 1, that in my version result split between pages 4 and 5. Furthermore, also the figure 5 should be paginate in a single page.

 Thank you for your comment. We have edited Table 1 and Figure 5 accordingly, so that they are all on one page.

  1. Moreover, always the figure 5, is both not mentioned and it is not discussed in the in the text. Figure 5 should be discussed in the text or delete.

Thank you for your comment. In line 502 we have mentioned Figure 5 in the text and it has been discussed (line 501-508).

  1. Finally, the conclusions are in accordance with the data discussed and with the aim of the document, however, the text from line 515 to line 530 is a repetition of the comment of the discussion, this part of the text should be rewritten.

Thank you for your comment. We have rewritten the part of Conclusions (lines 562, 566-569, 571-575, 578-580, 584-585, 588-591).

Reviewer 3 Report

Dear authors the manuscript you submitted represents an important aspect of the possibility of improving the health value of food products through the use of various sources of dietary fiber. However, I believe some peers need to be improved before publication. The introduction should be improved to better frame the aspects that will be treated in the manuscript, giving a greater scientific slant and concentrating the aims of the work at the end of the paragraph. Discussions need to be improved. In the tables it is necessary to insert / improve the caption in order to make them readable independently of the text of the manuscript. Furthermore, the significance must be verified; Table 2 shows in the caption that the comparisons are made between the means of the same column but, apart from the excessive lack of homogeneity of the results, it is not possible to verify the significance. Also in table 3 the same problem with the significance among the means   Best regards

Author Response

University of Life Sciences in Lublin

Faculty of Food Sciences and Biotechnology

Department of Milk Technology and Hydrocolloids

Skromna 8, 20-704 Lublin

Poland

Phone: +48 81 4623350

Fax:     +48 81 4623345

E-mail: bartosz.solowiej@up.lublin.pl

March 25, 2021,

Dear Reviewer,

Our manuscript entitled “The influence of dietary fibers on physicochemical properties of acid casein processed cheese sauces obtained with whey proteins and coconut oil or anhydrous milk fat” (Ref. No.: foods-1149884) has been revised and is being re-submitted for publication in Foods – Special Issue “New Frontiers in Dairy Technology and Hydrocolloids”

We have carefully considered each of the comments and made the appropriate revisions in the manuscript. An itemized list of our responses to each of the comments is included below.

Thank you for your kind attention.

Yours faithfully,

Bartosz Sołowiej

We have corrected our manuscript with regard to Reviewers’ comments:

Reviewer #1:  green color

Reviewer #2: blue color

Reviewer #3: yellow color

If Reviewers had the same comments – underline

Reviewer #3: Review of Manuscript Number: foods-1149884

Title: The influence of dietary fibers on physicochemical properties of acid casein processed cheese sauces obtained with whey proteins and coconut oil or anhydrous milk fat

Dear authors the manuscript you submitted represents an important aspect of the possibility of improving the health value of food products through the use of various sources of dietary fiber. However, I believe some peers need to be improved before publication.

Thank you very much for your opinion about the manuscript.

The introduction should be improved to better frame the aspects that will be treated in the manuscript, giving a greater scientific slant and concentrating the aims of the work at the end of the paragraph.

Thank you for your comment. We have concentrated the aims of the work in the end of this section (lines 103-106). Moreover, we have improved the Introduction section (lines 44-69).

References added to the Introduction section:

  1. SzafraÅ„ska, J.O.; SoÅ‚owiej, B.G. Cheese sauces: Characteristics of ingredients, manufacturing methods, microbiological and sensory aspects. J. Food Process Eng. 2020, 43, 1–14, doi:10.1111/jfpe.13364.
  2. Shinde, G.; Kumar, R.; Chauhan, S.K.; Shinde, G.; Subramanian, V.; Nadanasabapathi, S. Whey Proteins: A potential ingredient for food industry- A review. Asian J. Dairy Food Res. 2018, doi:10.18805/ajdfr.dr-1389.
  3. Sarode A.R., Sawale P.D., Khedkar C.D., Kalyankar S.D., P.R.D. Casein and caseinates. Methods of Manufacture. In The Encyclopedia of Food and Health; Caballero, B., Finglas, P., Toldrá, F., Ed.; Oxford: Academic Press, 2016; pp. 676–682.
  4. Wang, J.; Wang, X.; Li, J.; Chen, Y.; Yang, W.; Zhang, L. Effects of dietary coconut oil as a medium-chain fatty acid source on performance, carcass composition and serum lipids in male broilers. Asian-Australasian J. Anim. Sci. 2015, 28, 223–230, doi:10.5713/ajas.14.0328.
  5. Deen, A.; Visvanathan, R.; Wickramarachchi, D.; Marikkar, N.; Nammi, S.; Jayawardana, B.C.; Liyanage, R. Chemical composition and health benefits of coconut oil: an overview. J. Sci. Food Agric. 2020, 101, 2182–2193, doi:10.1002/jsfa.10870.
  6. Herrera-Meza, M.S.; Mendoza-López, M.R.; García-Barradas, O.; Sanchez-Otero, M.G.; Silva-Hernández, E.R.; Angulo, J.O.; Oliart-Ros, R.M. Dietary anhydrous milk fat naturally enriched with conjugated linoleic acid and vaccenic acid modify cardiovascular risk biomarkers in spontaneously hypertensive rats. Int. J. Food Sci. Nutr. 2013, 64, 575–586, doi:10.3109/09637486.2013.763908.
  7. Sołowiej, B.G.; Nastaj, M.; Szafrańska, J.O.; Muszyński, S.; Gustaw, W.; Tomczyńska-Mleko, M.; Mleko, S. Effect of emulsifying salts replacement with polymerised whey protein isolate on textural, rheological and melting properties of acid casein model processed cheeses. Int. Dairy J. 2020, 105, doi:10.1016/j.idairyj.2020.104694.
  8. Barber, T.M.; Kabisch, S.; Pfeiffer, A.F.H.; Weickert, M.O. The health benefits of dietary fibre. Nutrients 2020, 12, 1–17, doi:10.3390/nu12103209.

Discussions need to be improved. In the tables it is necessary to insert / improve the caption in order to make them readable independently of the text of the manuscript.

Thank you for your comment. We have improved the Discussion section in lines 303-306, 344-347, 358-361 and 389-393 we added additional information to clarify mentioned subject. 

Moreover, we have improved captions in Tables (lines: 395-396, 498-499, 527-528) and Figures (lines 215-217, 254-256, 299-301, 506-508).

References added to the Discussion section:

  1. SzafraÅ„ska, J.O.; SoÅ‚owiej, B.G. Effect of different fibres on texture, rheological and sensory properties of acid casein processed cheese sauces. Int. J. Food Sci. Technol. 2020, 55, 1971–1979, doi:10.1111/ijfs.1448541.

41. Dikeman, C.L.; Fahey, G.C. Viscosity as related to dietary fiber: A review. Crit. Rev. Food Sci. Nutr. 2006, 46, 649–663, doi:10.1080/10408390500511862.

46. Kinsella, J.E. Milk proteins: Physicochemical and functional properties. C R C Crit. Rev. Food Sci. Nutr. 1984, doi:10.1080/10408398409527401.

47. Hunt, J.A.; Dalgleish, D.G. Adsorption behaviour of whey protein isolate and caseinate in soya oil-in-water emulsions. Top. Catal. 1994, doi:10.1016/S0268-005X(09)80042-8.

Furthermore, the significance must be verified; Table 2 shows in the caption that the comparisons are made between the means of the same column but, apart from the excessive lack of homogeneity of the results, it is not possible to verify the significance. Also in table 3 the same problem with the significance among the means.

Thank you for your comment. Based on your comment, we did look into analysis to our study. We have performed additional post hoc analysis, applying Student-Newman-Keuls and Duncan test, which as being less conservative test are more likely to reveal significant differences between group means that Tukey’s HSD test. Both above mentioned tests have shown additional significant differences in groups. Thus, we have decided to proceed with this Duncan test, as it has greater statistical power than Student-Newman-Keuls. We thank you for this question, which made us look into our statistical analysis again.

Thank you very much.

Round 2

Reviewer 3 Report

Dear Authors
I have cheched your revised manuscript and the changes made improve the work. Unfortunately, I have some doubts about the last point "Furthermore, the significance must be verified; Table 2 shows in the caption that the comparisons are made between the means of the same column but, apart from the excessive lack of homogeneity of the results, it is not possible to verify the significance. Also in table 3 the same problem with the significance among the means. ".
Specifically, I was referring to the readability of the table and the significance of the differences between the averages reported superscripts; for example in the first column of table 2 the differences between the means are indicated with superscripts from "a" to "l" but "f and g" are missing. The same error is repeated in other columns.
I invite you to review and correct the significance between the differences in the averages in the tables, paying attention to the superscripts used.

Best regard

Author Response

University of Life Sciences in Lublin

Faculty of Food Sciences and Biotechnology

Department of Milk Technology and Hydrocolloids

Skromna 8, 20-704 Lublin

Poland

Phone: +48 81 4623350

Fax:     +48 81 4623345

E-mail: bartosz.solowiej@up.lublin.pl

March 28, 2021,

Dear Reviewer,

Our manuscript entitled “The influence of dietary fibers on physicochemical properties of acid casein processed cheese sauces obtained with whey proteins and coconut oil or anhydrous milk fat” (Ref. No.: foods-1149884 – Round 2) has been revised and is being re-submitted for publication in Foods – Special Issue “New Frontiers in Dairy Technology and Hydrocolloids”

We have carefully considered each of the comments and made the appropriate revisions in the manuscript. An itemized list of our responses to each of the comments is included below.

Thank you for your kind attention.

Yours faithfully,

Bartosz Sołowiej

We have corrected our manuscript with regard to Reviewers’ comments:

Reviewer #3: yellow color

Reviewer #3: Review of Manuscript Number: foods-1149884

Title: The influence of dietary fibers on physicochemical properties of acid casein processed cheese sauces obtained with whey proteins and coconut oil or anhydrous milk fat

Dear Authors
I have cheched your revised manuscript and the changes made improve the work. Unfortunately, I have some doubts about the last point "Furthermore, the significance must be verified; Table 2 shows in the caption that the comparisons are made between the means of the same column but, apart from the excessive lack of homogeneity of the results, it is not possible to verify the significance. Also in table 3 the same problem with the significance among the means. ".
Specifically, I was referring to the readability of the table and the significance of the differences between the averages reported superscripts; for example in the first column of table 2 the differences between the means are indicated with superscripts from "a" to "l" but "f and g" are missing. The same error is repeated in other columns.
I invite you to review and correct the significance between the differences in the averages in the tables, paying attention to the superscripts used.

Thank you very much. We have modified table captions (Table 2, 3 and 4).

Statistical analysis in our article was performed for all fibers (AF, CF, PF, BF – (columns)) and concentrations (1, 2, 3, 4% - (rows)) within 2 fats/oils (OCO and AMF - (columns)). However, the caption for the table was incorrectly assigned (only columns). For this reason, a correction was made to the table captions and not to the letters themselves. We hope that in its current form it does not raise objections.

Table 2

It was: a-t Means in the same column with different superscripts are significantly different (P < 0.05, Duncan test).

It is: a-t Means within columns and rows for each parameter (G′, G″, tan (δ) or yield stress) with different superscripts are significantly different (P < 0.05, Duncan’s test).

Table 3

It was: a-r Means in the same column with different superscripts are significantly different (P < 0.05, Duncan test).

It is: a-r Means within columns and rows for each parameter (L*, a*, b*, Chroma or Hue) with different superscripts are significantly different (P < 0.05, Duncan’s test).

Table 4

It was: a-o Means in the same column with different superscripts are significantly different (P < 0.05, Duncan HSD test).

It is: a-o Means within columns and rows for each parameter (Density or Water activity) with different superscripts are significantly different (P < 0.05, Duncan’s test).
